# Inclusion Complexes of 3,4-Ethylenedioxythiophene with Per-Modified β- and γ-Cyclodextrins

**DOI:** 10.3390/molecules28083404

**Published:** 2023-04-12

**Authors:** Aurica Farcas, Ana-Maria Resmerita, Mihaela Balan-Porcarasu, Corneliu Cojocaru, Cristian Peptu, Ion Sava

**Affiliations:** “Petru Poni” Institute of Macromolecular Chemistry, 700487 Iasi, Romania; resmerita.ana@icmpp.ro (A.-M.R.); mihaela.balan@icmpp.ro (M.B.-P.); cojocaru.corneliu@icmpp.ro (C.C.); cristian_peptu@yahoo.com (C.P.)

**Keywords:** 2,3,6-tri-*O*-methyl β- or γ-cyclodextrin, 3,4-ethylenedioxythiophene, supramolecular encapsulation, molecular docking, matrix-assisted laser desorption ionization mass spectroscopy, differential scanning calorimetry

## Abstract

Herein, we report the synthesis of inclusion complexes (ICs) based on 3,4-ethylenedioxythiophene (EDOT) with permethylated β-cyclodextrins (TMe-βCD) and permethylated γ-cyclodextrins (TMe-γCD) host molecules. To prove the synthesis of such ICs, molecular docking simulation, UV-vis titrations in water, ^1^H-NMR, and H-H ROESY, as well as matrix-assisted laser desorption ionization mass spectroscopy (MALDI TOF MS) and thermogravimetric analysis (TGA) were carried out on each of the EDOT∙TMe-βCD and EDOT∙TMe-γCD samples. The results of computational investigations reveal the occurrence of hydrophobic interactions, which contribute to the insertion of the EDOT guest inside the macrocyclic cavities and a better binding of the neutral EDOT to TMe-βCD. The H-H ROESY spectra show correlation peaks between H-3 and H-5 of hosts and the protons of the guest EDOT, suggesting that the EDOT molecule is included inside the cavities. The MALDI TOF MS analysis of the EDOT∙TMe-βCD solutions clearly reveals the presence of MS peaks corresponding to sodium adducts of the species associated with the complex formation. The IC preparation shows remarkable improvements in the physical properties of EDOT, rendering it a plausible alternative to increasing its aqueous solubility and thermal stability.

## 1. Introduction

Increasing demand for high-performance applications of conjugated polyrotaxanes requires a rational approach in the selection of host and guest components as the first step for achieving improvements of their photophysical properties. These supramolecular architectures diminish the intermolecular interactions and offer a way to construct a better protective sheath around the conjugated chains. Over the past two decades, the scientific interest has focused on the possibility of controlling intermolecular interactions by insulating various π-conjugated monomers, oligomers, or polymers to preserve the photophysical characteristics towards an organic bioelectronic application [1]. Among the various conjugated monomers, the EDOT compound is widely recognized as a material of interest. This is justified by its chemical structure, where the oxygens play an important role in stabilizing the positive charges in the PEDOT chains, thereby offering great opportunities for improving the interactions with biological entities [1,2,3,4,5,6]. Molecular encapsulation with the formation of inclusion complexes (ICs), which involves the presence of macrocyclic molecules (hosts) threaded over different monomers/polymers cores (guests) without any covalent bonds between them, has been extensively studied as a significant topic in both chemistry and biology [6,7,8,9,10]. A wide variety of host molecules have the ability to bind various guests into their interior cavity [6,7,8,9,10,11,12,13,14,15,16]. In general, hydrophobic, neutral, or polar molecules are able to form such ICs with native cyclodextrins (CDs) and their derivatives, with the resulting organized structures relevant for applications in various fields [2,3,4,5,7,8,11,13,14,17].

In order to further investigate EDOT-based polymers, we herein present novel results where EDOT units encapsulated into permodified CDs were characterized in terms of structure, stability of the inclusion complexes, and association constant. The insight gained from studying these compounds is a first step towards generating complex functional materials. We focused on TMe-βCD and TMe-γCD hosts for the synthesis of EDOT∙TMe-βCD and EDOT∙TMe-γCD in order to overcome the low water solubility of EDOT, as solubility is a necessary property for a convenient and efficient synthesis of PEDOT pseudo- and polyrotaxane architectures. We selected TMe-βCD and TMe-γCD host molecules on the basis of reported improvements of photophysical and transport properties of PEDOT after EDOT complexation into the TMe-βCD cavity [17]. Our results provide, for the first time to the best of our knowledge, novel findings and a better understanding of the interactions of the neutral EDOT monomer with TMe-βCD and TMe-γCD, insight which is instrumental for the field of nanobiotechnology applications.

Therefore, we report the preparation of EDOT∙TMe-βCD and EDOT∙TMe-γCD and systematically investigate the ability of TMe-βCD and TMe-γCD to encapsulate the neutral EDOT monomer. Molecular docking simulation and MALDI MS techniques validate the complexation ability of TMe-βCD and TMe-γCD host molecules towards EDOT, thus leading to EDOT∙TMe-βCD and EDOT∙TMe-γCD ICs formation. The complexation ability was further analyzed using UV-vis titrations in water, ^1^H-NMR, and H-H ROESY (two-dimensional rotating frame overhauser enhancement spectroscopy). Further information is provided by TGA analysis.

## 2. Results and Discussion

### Synthesis and Characterization

IC preparation consists of threading of the macrocyclic host onto the monomer/polymer chains (guest) without any covalent bonds between them. The synthesis of such complex structures is based on the molecular recognition principle and is a result of the cooperation of various noncovalent interactions, such as hydrophobic, electrostatic, or van der Waals. Native CDs are by far the most intensively investigated host molecules in the synthesis of such ICs. Through the chemical modification of the native CDs, their hydrophilic character decreases, and this property enables the permodified CDs derivatives to bind neutral guests into their interior cavity.

The macrocycles TMe-βCD and TMe-γCD were prepared according to previously reported procedures [18,19]. The chemical structures of TMe-βCD and TMe-γCD were characterized by ^1^H- and ^13^C-NMR spectroscopy (Appendix A). The assignment of the peaks confirms the synthesis of TMe-βCD or TMe-γCD, in agreement with previously reported data [18,19,20].

The synthesis of EDOT∙TMe-βCD and EDOT∙TMe-γCD ICs was carried out in water/acetone 4/1 *v*/*v* using a 1:1 molar ratio of macrocycles and the EDOT monomer according to the synthetic route shown in Figure 1. The EDOT was, therefore, solubilized in a minimal volume of acetone and dropwise added to a saturated water solution of TMe-βCD or TMe-γCD (Note in Section 3).

As these ICs are largely unexplored materials, we confirmed their formation using multiple techniques. To simulate the inclusion complex systems between the TMe-βCD and TMe-γCD macrocycles and the neutral EDOT guest, molecular docking simulation was performed. Molecular docking of TMe-βCD and TMe-γCD molecules and EDOT was carried out using the AutoDock-(LGA) searching algorithm [21], encompassed in the YASARA-Structure software (v.20.8.23) [22,23]. In this investigation, the TMe-βCD and TMe-γCD (hosts) were treated as rigid bodies, whereas the EDOT was treated as a flexible part. Next, both the rigid and flexible components were assembled according to the automatic docking algorithm at the level of molecular mechanics theory using the YASARA force field (Figure 1 and Figure 2).

The computations were performed using a number of 100 docking runs followed by cluster analysis. All computations were accomplished using a Dell Precision Workstation T7910. The global docking simulation for the EDOT∙TMe-βCD and EDOT∙TMe-γCD host-guest systems was achieved using a simulation cell of size 36 × 36 × 36 Å^3^ and 0.375 Å grid spacing. These spatial parameters were sufficient to enclose the molecular volumes of both complexes. The binding energy (*E_b_*) estimated by scoring function was equal to −3.94 kcal∙mol^−1^, and the dissociation constant (*K_d_*) was found to be 1.30 mM for EDOT∙TMe-βCD. The results of computational investigations for EDOT∙TMe-γCD indicated the binding of the neutral EDOT to TMe-γCD, with the values of *E_b_* and *K_d_* −4.09 kcal∙mol^−1^ and 1.0 mM. The lower *E_b_* and *K_d_* values for EDOT∙TMe-γCD compared with those of EDOT∙TMe-βCD could be attributed to the loose binding of EDOT inside the larger cavity volume of the TMe-γCD. Moreover, the computational results revealed the existence of hydrophobic interactions, illustrated as solid green lines, which are attractive forces and contribute to the insertion of the EDOT guest inside both macrocyclic cavities. In order to further validate that the TMe-βCD and TMe-γCD macrocycles are able to recognize the neutral EDOT guest, UV-vis titrations were also performed. The stability constant (*K_s_*) was evaluated by UV−vis absorption measurements in water with an increasing concentration of TMe-βCD or TMe-γCD macrocycles and the fitting according to a 1:1 host–guest complexation stoichiometry (Appendix A). *K_s_* values of ~928 (±80) and 764 (±60) M^−1^ were obtained for the formation of EDOT∙TMe-βCD and EDOT∙TMe-γCD, respectively. The data are similar to other *K_s_* values measured in water of EDOT ICs previously reported [6,24] and in agreement with the computational results described above. The main driving forces involved in IC formation are the hydrophobic–hydrophobic interactions between the neutral EDOT guest and the TMe-βCD or TMe-γCD [25]. The chemical structures of EDOT∙TMe-βCD and EDOT∙TMe-γCD were determined by combining FT-IR and ^1^H-NMR spectroscopy. FT-IR spectrum of EDOT∙TMe-βCD revealed the characteristic bands of EDOT [26] and those of TMe-βCD (Appendix A). The vibrations at 2922 cm^−1^, 1516, and 1344 cm^−1^ are attributed to the stretching modes of the dioxyethylene and C=C and C–C in the thiophene ring. The vibration modes of the C–S bond in the thiophene ring can be seen at 980, 841, and 689 cm^−1^. The bands at 1202 and 1090 cm^−1^ are assigned to the stretching modes of the ethylenedioxy group. Besides the bands attributed to the EDOT, in the FT-IR spectrum, additional bands appeared located at 1073, 1061, 571, 519, and 434 cm^−1^, corresponding to the presence of TMe-βCD [27].

Further insight was provided by ^1^H-NMR spectroscopy in D_2_O (Figure 3). It should be noted that, due to the insolubility of EDOT in water, ICs formed with permodified CDs were never confirmed using NMR spectroscopy techniques. In general, if a guest molecule is located within the CD cavity, the hydrogen atoms on the cavity’s inner surface (H-3 and H-5) will be shielded by the guest [28]. The ^1^H-NMR spectra of the EDOT∙TMe-βCD and EDOT∙TMe-γCD clearly denoted chemical shift displacements between 0.03 and 0.08 ppm for all the protons of both host molecules (Appendix A). The largest chemical shift variations were observed for the protons located inside the hydrophobic cavity (H-3 and H-5). These variations occurred because the water from inside the host cavity was replaced by the hydrophobic EDOT molecule, exposing the inner cavity protons to the aromatic ring current of the EDOT residue. These changes in the spectra are indicative of EDOT encapsulation inside the macrocyclic cavities. The H-3 and H-5 protons of EDOT∙TMe-βCD showed shift changes of 0.08 and 0.05 ppm, respectively, suggesting that the EDOT molecule entered through the secondary side (Figure 3). Given that H-5 is located closer to the primary rim of CD derivatives, the interaction between guest and host system suggests that the EDOT molecule entered deeply inside the TMe-βCD cavity [29]. These observations are in agreement with the differences between the conformation of native CDs and permodified derivatives, which is essential for the molecular recognition of the EDOT by TMe-βCD or TMe-γCD species. In these permodified macrocycles, many methyloxyls instead of hydroxyls favor a more extended cavity and torus rims [30]. According to the ^1^H-NMR data, we, therefore, conclude that the EDOT was included in the central cavity.

In order to obtain a more detailed structural assessment of the EDOT∙TMe-βCD and EDOT∙TMe-γCD samples, two-dimensional H-H ROESY experiments were also applied, taking into account the spatial proximity (below 5 Å) between EDOT and both permodified CD protons (Figure 4 and Appendix A).

The H-H ROESY spectrum for EDOT∙TMe-βCD showed correlation peaks between H-3 and H-5 and the aromatic CH protons of EDOT, suggesting that the EDOT molecule was included with the thiophene residue deep inside the cavity (Figure 4). The H-H ROESY spectrum did not show correlation peaks between EDOT and the protons from the outside cavity of macrocycles. The chemical shift displacements of H-1, H-2, H-4, and the -OMe groups could be ascribed to conformational changes of the glucopyranose units that occur upon complexation [31,32].

The above studies suggest that EDOT encapsulation into TMe-βCD and TMe-γCD is clearly possible and support the observation that EDOT∙TMe-βCD and EDOT∙TMe-γCD were synthesized.

The formation of the inclusion complex between TMe-βCD host and EDOT guest molecule was further investigated using mass spectrometry (MS). Generally, the formation of ICs is studied using electrospray mass spectrometry (ESI-MS) due to the specific ionization conditions through solvent depletion, which promotes the host–guest physical association in the presence of CD cyclic compounds. However, such analysis may induce some errors regarding the presence of such ICs before or after the electrospray ionization process [33]. On the other hand, MALDI MS ionization provides rather energetic conditions that may lead to inclusion complexes dissociation and, therefore, is less used for complexation studies [34]. Nevertheless, the formation of the EDOT∙CB7 ICs was previously shown using MALDI MS/MS [6]. The direct MALDI TOF MS analysis of the EDOT∙TMe-βCD solutions revealed the presence of small MS peaks corresponding to sodium adducts of the species associated with the complex formation. However, because of the relatively weak physical interactions and reduced ionization efficiency of the bulky complexes, the associated MALDI TOF MS complex peaks presented a low intensity, which increased the uncertainty of the MS identification. Moreover, the host–guest association between the EDOT and TMe-γCD could not be observed. We, therefore, decided to perform a fragmentation experiment for the EDOT∙TMe-βCD in an attempt to study the nature of the observed MS peaks. The parent ions observed at *m*/*z* = 1593 (*m*/*z* = 1418 (TMe-βCD) + 142 (EDOT) + 23 (Na)) were associated with the presence of the EDOT∙TMe-βCD, which was further subjected to laser-induced dissociation fragmentation (LID MS/MS) in a TOF/TOF MALDI MS setup, operated in LIFT mode (Figure 5 and Appendix A).

The MS/MS spectrum revealed that the main fragmentation pathway was represented by the dissociation of the EDOT∙TMe-βCD parent ion through the neutral loss of the EDOT moiety (Δ*m*/*z* = 142 Da) and the formation of the Na-charged fragment at *m*/*z* = 1451, associated with TMe-βCD. Thus, the fragmentation experiment confirmed that the observed MALDI MS peak was correctly associated with the presence of the noncovalent EDOT∙TMe-βCD complex.

The thermal analysis provided further information about the IC formation. TGA was performed on pure TMe-γCD, EDOT∙TMe-γCD, and the physical mixture between host and guest (Figure 6). The thermogram of TMe-γCD exhibited two weight losses at 318 and 377 °C. The EDOT∙TMe-γCD underwent weight losses at 157 and 336 °C, losing 87% of its original weight at 380 °C. To confirm the formation of the EDOT∙TMe-γCD, TGA of the physical mixture between the TMe-γCD and EDOT was also performed. The TGA curve showed that the first weight loss for the physical mixture started at 143 °C in comparison with the EDOT∙TMe-γCD, which started at 157 °C (Table 1). However, the formation of EDOT∙TMe-γCD changed the thermal properties of TMe-γCD and EDOT. In addition, the second decomposition for EDOT∙TMe-γCD was at approximately 336 °C, whereas for the TMe-γCD, it was identified at 377 °C. This phenomenon suggests that these two components in the EDOT∙TMe-γCD interacted, and its formation decreased the thermal stability of the TMe-γCD molecule. These last assessments unambiguously revealed that the thermal stability in the physical mixtures could be associated with the lack of host–guest interactions between the EDOT and macrocycle hosts [35,36].

At the same time, it can be observed that the encapsulation of EDOT into the TMe-βCD cavity had a greater impact on the first weight loss in the case of EDOT∙TMe-βCD as well as the physical mixture (Appendix A). This observation strongly suggests that the formation of EDOT∙TMe-βCD slightly decreased the TMe-βCD stability (Table 2).

## 3. Materials and Methods

### 3.1. Synthesis and Characterization

EDOT was purchased from Sigma-Aldrich and purified before use by vacuum distillation. Sodium hydride (NaH) 60% dispersion in mineral oil and iodomethane were acquired from Sigma-Aldrich. β-CD and γ-CD were purchased from CycloLab Ltd. (Budapest, Hungary), recrystallized twice from water, and then dried in vacuum at 95 °C for 24 h prior to use. For MALDI MS analysis, the matrix (*trans*-2-(3-(4-tert-butylphenyl)-2-methyl-2-propenylidene)) malononitrile-DCTB and sodium iodide (NaI) were purchased from Sigma Aldrich (Saint Louis, MO, USA). Ultrapure water and all other solvents were purchased from commercial sources (Sigma-Aldrich, Fischer (Ried im Innkreis, Austria)) and used as received.

### 3.2. Characterization

The FT-IR (KBr pellets) spectra were obtained on a Bruker Vertex 70 spectrophotometer. The NMR spectra were recorded on Bruker Avance Neo 400 MHz spectrometer equipped with a 5 mm inverse-detection z-gradient multinuclear probe using the standard parameter sets provided by Bruker. H-H ROESY experiments were recorded with water signal suppression, with a mixing time of 200 milliseconds. ^1^H and ^13^C-NMR spectra were recorded at room temperature and were referenced on the solvent residual peak (^1^H: 4.80 ppm for D_2_O and 7.26 ppm for CDCl_3_; ^13^C: 77.0 ppm for CDCl_3_). The molecular docking simulations were conducted using the AutoDock-(LGA) searching algorithm [21], encompassed in the YASARA structure software package [22,23]. Mass Spectrometry: MALDI MS analysis was performed using a RapifleX MALDI TOF TOF MS instrument (Bruker, Bremen, Germany). FlexControl 4.0 and FlexAnalysis 4.0 software (Bruker (Bremen, Germany)) were used to control the instrument and process the MS and MS/MS spectra. The samples for MS analysis were prepared by dissolving in dry THF (10 mg∙mL^−1^) and mixed using a Vortex-Genie 2 device. The DCTB matrix was prepared in THF at a concentration of 10 mg∙mL^−1^, while the NaI concentration in THF solution was 5 mg∙mL^−1^. The samples were applied on the MALDI steel plate using the dried droplet method: 20 μL of matrix solution was mixed with 1 μL of NaI and 2 μL of sample solution, and 1 μL from this mixture was deposited on the ground steel plate. The spectra were acquired in the positive reflectron mode, and the laser ionization power was adjusted slightly above the threshold to produce consistent MS signals. The MS calibration was performed using poly(ethylene glycol) standards applied to the MALDI MS target. The MALDI MS/MS fragmentation experiments were performed in LIFT mode using a Bruker standard fragmentation method. All the instrument parameters employed for the MS/MS process were controlled by the standard MS/MS method provided by the instrument producer. The complete isotopic profile of the parent ion was isolated. The thermogravimetric analysis was carried out on a Mettler Toledo TGA/SDTA 851e equipment (Mettler Toledo, Greifensee, Switzerland) under constant N_2_ flow (20 cm^3^∙min^−1^) and heating rate of 10 °C min^−1^ from 25 to 800 °C. The TG curves were processed with Mettler Toledo STAR^e^ software (Version 9.10, Giessen, Germany).

### 3.3. Synthesis of TMe-βCD and TMe-γCD

TMe-βCD and TMe-γCD were synthesized by adding a suspension of NaH to a βCD or γCD solutions in dry DMF and stirring the mixture for 8 h. Then, the mixture was cooled to 0 °C, and an excess of iodomethane was added dropwise. The resulting reaction mixture was extracted with CHCl_3_ three times. The combined organic layer was washed with water until pH = 7, dried over MgSO_4_, concentrated in vacuum, and recrystallized from ethylacetate/heptan 1/4 *v*/*v*. TMe-βCD was obtained as a white powder in a 68% yield, whereas TMe-γCD was only in a 54% yield.
^1.^ H-NMR (D_2_O, 400 MHz), δ (ppm): 5.29 (d, *J* = 3.5 Hz, 7H, H1), 3.88–3.85 (m, 14H, H5, H6a), 3.76 (t, *J* = 8.9 Hz, 7H, H4), 3.72–3.65 (m, 14H, H3, H6b), 3.61 (s, 21H, C2-OCH_3_), 3.52 (s, 21H, C3-OCH_3_), 3.39 (s, 21H, C6-OCH_3_), 3.36 (dd, *J* = 3.5 Hz, *J* = 9.6 Hz, 7H, H2).
^13.^ C NMR (D_2_O, 100 MHz), δ (ppm): 97.0 (C1), 80.9 (C3), 80.0 (C2), 77.0 (C4), 70.7 (C6), 70.4 (C5), 57.9 (C2-OCH_3_), 58.4 (C6-OCH_3_), 58.1 (C3-OCH_3_).


### 3.4. Synthesis of EDOT∙TMe-βCD and EDOT∙TMe-γCD

For the synthesis of EDOT∙TMe-βCD, freshly distilled EDOT (570 mg, 4.0 mmol) solubilized in 3 mL of acetone was added to 15 mL of clear aqueous solution of TMe-βCD (5.712 g, 4.0 mmol) followed by vigorous stirring at ambient temperature for 48 h under N_2_ to produce a clear solution. The solvents were removed by freeze-drying, and the EDOT∙TMe-βCD complex was obtained as a white powder.
^1.^ H-NMR (D_2_O, 400 MHz), δ (ppm): 6.50 (s, Ar-H from EDOT), 5.25 (d, *J* = 3.5 Hz, H1), 4.26 (s, CH_2_ from EDOT), 3.83–3.81 (m, H5, H6a), 3.72 (t, *J* = 9.0 Hz, H4), 3.64–3.60 (m, H3, H6b, C2-OCH_3_), 3.51 (s, C3-OCH_3_), 3.39 (s, C6-OCH_3_), 3.31 (dd, *J* = 3.4 Hz, *J* = 9.9 Hz, H2).

The synthesis of the inclusion complex EDOT·TMe-γCD was performed under similar experimental conditions as those used for the preparation of the EDOT·TMe-βCD inclusion complex, except (6.422 g, 0.4 mmol) of TMe-γCD was used instead of TMe-βCD.
^1.^ H-NMR (D_2_O, 400 MHz), δ (ppm): 6.51 (s, Ar-H from EDOT) 5.26 (d, *J* = 3.4 Hz, H1), 4.26 (s, CH_2_ from EDOT), 3.84–3.81 (m, H5, H6a), 3.73 (t, *J* = 9.0 Hz, H4), 3.66–3.62 (m, H3, H6b, C2-OCH_3_), 3.52 (s, C3-OCH_3_), 3.39 (s, C6-OCH_3_), 3.32 (dd, *J* = 3.4 Hz, *J* = 9.9 Hz, H2).

## 4. Conclusions

In conclusion, the results described here demonstrate the binding ability of EDOT to TMe-βCD and TMe-γCD host molecules. EDOT∙TMe-βCD and EDOT∙TMe-γCD formation led to distinct improvements of EDOT physical properties. It was found that IC formation significantly enhanced the aqueous solubility and thermal stability of EDOT. According to the molecular docking simulation, the EDOT∙TMe-βCD and EDOT∙TMe-γCD were stabilized by hydrophobic and electrostatic forces. The ^1^H-NMR chemical shift changes provide further validation for the formation of EDOT∙TMe-βCD and EDOT∙TMe-γCD. According to the H-H ROESY experiments, the EDOT is included in the central cavity of TMe-βCD or TMe-γCD host molecules. TGA analysis showed that the formation of EDOT∙TMe-βCD and EDOT∙TMe-γCD changed the thermal properties of both guest and host molecules. With this study, we hope to provide further insights and accurately quantify the effect of permodified CDs encapsulation on the physical properties of encapsulated EDOT monomer. According to these investigations, we can conclude that EDOT∙TMe-βCD and EDOT∙TMe-γCD ICs are better organized systems with improved thermal stability and water solubility than those of non-encapsulated EDOT monomer. This study can be extended to build new supramolecular compounds of interest.

## Data Availability

All data that support the findings of this study are incorporated in the main manuscript and in the Appendix A of this article. No additional data are available for sharing.

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
