# Peer review of "Inclusion Complexes of 3,4-Ethylenedioxythiophene with Per-Modified β- and γ-Cyclodextrins"

_molecules, 2023, doi:10.3390/molecules28083404_

Round 1

Reviewer 1 Report

Farcas et al. (molecules-2355882) studied interesting adducts formed by two hydrophobic components (one is a derivative of thiophene, another is a permetylated cyclodextrin). They appear to have properly characterized these adducts experimentally, and even have performed a low-level modeling of the complexes. I recommend their paper for publication once the following five points are addressed:

1. I think some recent references should appear in Intro, e.g.,
Percastegui, E. G.; Ronson, T. K.; Nitschke, J. R. Desing and Applications of Water-Soluble Coordination Cages. Chem. Rev. 2020, 120, 13480–13544. DOI: 10.1021/acs.chemrev.0c00672

2. Abbreviations should be introduced not only in Abstract. I think their list could be useful.

3. Is the particular orientation of EDOT inside a host in Figure 4 just an illustrative look? Is it based on some data or assumptions? This should be stated in the paper.

4. I think the fitting procedure used to estimate the binding constant should be described in more detail (by the way in Figure S5 the data-point most to the right looks like an outlier).

5. I wished the authors commented on a possible diffusion of the guest. I think they should at least mention diffusion-ordered NMR spectroscopy (DOSY) could've been used.

The paper contains mistakes like

... our interest is being made in this direction ...

at l. 53, but they can be handled by the MDPI language editors.

Author Response

We would like to thank the editor and reviewer for the constructive and valuable comments and suggestions. We believe the overall readability and English language have been improved in this revised form of the manuscript, as recommended. The answers and clarifications requested by the reviewer are now included in the main text in red.

Comments from the Reviewer #1:

  1. I think some recent references should appear in Intro, e.g.,
    Percastegui, E. G.; Ronson, T. K.; Nitschke, J. R. Desing and Applications of Water-Soluble Coordination Cages. Chem. Rev. 2020, 120, 13480–13544. DOI: 10.1021/acs.chemrev.0c00672

Answer Q 1: We agree with the reviewer and have now updated the Introduction to include the references recommended by Reviewers 1 &2(page 1, bottom).

Q 2: Abbreviations should be introduced not only in Abstract. I think their list could be useful.

Answer Q 2: This is now been corrected i.e. abbreviations are now described when used for the first time in the main text.

Q 3:  Is the particular orientation of EDOT inside a host in Figure 4 just an illustrative look? Is it based on some data or assumptions? This should be stated in the paper.

Answer Q 3: We agree with the reviewer and thank them for pointing this out. The structure depicted in Figure 4 is the proposed inclusion geometry of EDOT∙TMe-βCD based on the correlation peaks from the ROESY spectrum. This was clarified in the revised manuscript in the NMR characterization section.

Q 4:  I think the fitting procedure used to estimate the binding constant should be described in more detail (by the way in Figure S5 the data-point most to the right looks like an outlier).

Answer Q 4: We appreciate the validity of this question. The complexation ability (Ks) of both macrocycles with EDOT was proved by UV-vis titrations in water. To measure the Ks, the EDOT was dispersed in water by vortex stirring. Then the Ks were determined by measuring the absolute optical density variation of EDOT dispersion in water with increasing concentration of macrocycles taking advantage of the ability of these host molecules to encapsulate and solubilize hydrophobic/aromatic rings. We recognize that a minor error could have occurred due to one point potentially being an outlier but we believe such an error is not essential for our studies. The discussion is not included in the Ms.

Q 5:  I wished the authors commented on a possible diffusion of the guest. I think they should at least mention diffusion-ordered NMR spectroscopy (DOSY) could've been used.

Answer Q 5: We thank the reviewer for the suggestion. The purpose of the paper is to report on the synthesis and structural characterization of EDOT∙TMeCDs. Of course, there are more studies that can be performed on inclusion complexes, diffusion studies included. Unfortunately, our NMR laboratory does not possess the technical setup required for performing DOSY experiments. This type of experiments cannot be performed on standard NMR spectrometers (a special module that is available for purchase from Bruker is needed). Since we cannot provide DOSY experiments, we cannot comment on the possible diffusion of the guest but we believe our study and characterizations are solid without it.

Comments on the Quality of English Language

The paper contains mistakes like

... our interest is being made in this direction ...

We have taken the reviewer’s comments on board and have tried to improve the readability throughout the Ms. This particular mistake was corrected as follows:

We focused on TMe-βCD and TMe-γCD hosts for the synthesis of EDOT∙TMe-βCD and EDOT∙TMe-γCD in order to overcome the low water solubility of EDOT as solubility is a necessary property for a convenient and efficient synthesis of PEDOT pseudo- and polyrotaxane architectures.

We hope that our answers help to clarify the reviewers’ comments.

Reviewer 2 Report

This manuscript titled "Inclusion Complexes of 3,4-Ethylenedioxythiophene with Per-modified β- and γ-Cyclodextrins" provides a comprehensive overview of the synthesis and characterization of EDOT inclusion complexes with permodified cyclodextrins (CDs). The authors use various analytical techniques, including 1H-NMR spectroscopy, H-H ROESY experiments, mass spectrometry, and thermal analysis, to provide evidence for the formation of these complexes.The authors also provide a discussion of the implications of their findings concerning the structure of the complexes.
This manuscript is well-written and provides a thorough overview of the synthesis and characterization of EDOT inclusion complexes with permodified CDs. However, some areas could benefit from significant revision. For example,  the discussion section could include further implications of the authors' findings.
First, the authors should provide more detail in discussing the 1H-NMR data, particularly concerning the chemical shift displacements of H-3 and H-5 protons. Additionally, the authors should include a more thorough description of the MALDI MS analysis, including the parameters used for the fragmentation experiment.
Finally, the authors should provide more detail on how their results could be used to build new supramolecular compounds.
With these revisions, this manuscript could be improved and provide an even better overview of this area of research. For example, it is essential to apply cyclodextrin to drug delivery systems 10.1016/j.molliq.2022.119548 and 10.1016/j.jiec.2021.10.028 or for CO2 capture 10.3390/molecules27217375. Please discuss this in the introduction or conclusion.

Scientific sound corrections (not essential, but will help make your manuscript clear to understand):
"The knowledge gained on this complex compound is a first step toward attaining a greater understanding of the properties and subsequently to generate complex functional materials." --> "The knowledge gained on this complex compound is a first step toward understanding the properties better and generating complex functional materials."
"The main driving forces involved in the ICs formation are the hydrophobic-hydrophobic interactions between the
neutral EDOT guest and the TMe-βCD or TMe-γCD as previously was reported [25]. " --> "The main driving forces
involved in the formation of ICs are the hydrophobic-hydrophobic interactions between the neutral EDOT guest and the TMe-βCD or TMe-γCD as previously reported [25]."
"Mostly, if a guest molecule is located within the CD cavity, the hydrogen atoms located on the inner surface of the cavity (H-3 and H-5) will be shielded by the guest [28]." --> "Mostly, if a guest molecule is located within the CD cavity, the hydrogen atoms on the cavity's inner surface (H-3 and H-5) will be shielded by the guest [28]."
"In these permodified macrocycles the presence of many methyloxyls instead of hydroxyls favors a more extended cavity and torus rims [30]." --> "In these permodified macrocycles, many methyloxyls instead of hydroxyls favor a more extended cavity and torus rims [30]."
"(Figure 4 and Figure S9 in the in the Supplementary Materials)." --> "(Figure 4 and Figure S9 in the Supplementary Materials)."
"Further information about the ICs formation is provided by the thermal analysis." --> "The thermal analysis provides further information about the ICs formation."
"at 157 and 336 oC and lost 87 %" --> "at 157 and 336 oC, losing 87 %"
"This phenomenon suggests that that these two" --> "This phenomenon suggests that these two"
"β-CD and γ-CD were purchased from CycloLab Ltd. and recrystallized twice from water and then dried in vacuum at 95 °C for 24 h prior to use." --> "β-CD and γ-CD were purchased from CycloLab Ltd., recrystallized twice from water, and then dried in vacuum at 95 °C for 24 h before use."
"The 1 H-NMR chemical shift changes provide the indication for the formation" --> "The 1 H-NMR chemical shift changes indicate the formation"

Style corrections:

"chemical structure where the oxygens play an important role" --> "chemical structure where the oxygens play an essential role"
"EDOT and the protons from the outside cavity of macrocycles" --> "EDOT and the protons from the external cavity of macrocycles"
"TMe-βCD cavity has a greater impact on the" --> "TMe-βCD cavity has a more significant impact on the"
The full isotopic profile of the parent ion was isolated.  --> "The complete isotopic profile of the parent ion was isolated."

Misprintes:

"showed remarkably" --> "showed remarkable" and "of physical" --> "in the physical"
"inclusion complex systems" --> "inclusion of complex systems"
"titrations was also performed" --> "titrations were also performed."
"two dimensional" --> "two-dimensional"
"As a result from the above studies" --> "As a result of the above studies"
"small MS peaks corresponding to " --> "small MS peaks correspond"

Author Response

We would like to thank the editor and reviewers for the constructive and valuable comments and suggestions. We believe the overall readability and English language have been improved in this revised form of the manuscript, as recommended. The answers and clarifications requested by the reviewers are now included in the main text in red.

Comments from the Reviewer #2:

Q1: First, the authors should provide more detail in discussing the 1H-NMR data, particularly concerning the chemical shift displacements of H-3 and H-5 protons.

Answer Q1:

This explanation has now been included in the revised manuscript.

Q2: Additionally, the authors should include a more thorough description of the MALDI MS analysis, including the parameters used for the fragmentation experiment.

Answer Q2: Generally, the LIFT TOF/TOF mass spectrometer consists of a gridless MALDI ion source with delayed extraction, a high-resolution timed ion selector, a "lift" device for raising the potential energy of the ions, a further velocity focusing stage with subsequent post-acceleration, a post lift metastable suppressor, a griddles space-angle and energy focusing reflector, and fast ion detectors for the linear and reflector mode, as described elsewhere [Suckau, D., Resemann, A., Schuerenberg, M. et al. A novel MALDI LIFT-TOF/TOF mass spectrometer for proteomics. Anal Bioanal Chem 376, 952-965 (2003). https://doi.org/10.1007/s00216-003-2057-0]. In this study, the MS/MS fragmentation was operated in LIFT mode and all the instrument physical parameters employed for the MS/MS process were controlled by the standard MS/MS method provided by the instrument producer, without any significant deviations. Therefore, we consider that the MS/MS experiment in our work was accurately described (as below). These explanations were not included in the revised manuscript.

The text was modified accordingly (red).

Mass Spectrometry: MALDI MS analysis was performed using a RapifleX MALDI TOF TOF MS instrument (Bruker, Bremen, Germany). FlexControl 4.0 and FlexAnalysis 4.0 software (Bruker) were used to control the instrument and process the MS and MS/MS spectra. The samples for MS analysis were prepared by dissolving in dry THF (10 mg∙mL-1) and mixed using a Vortex-Genie 2 device. The DCTB matrix was prepared in THF at a concentration of 10 mg∙mL-1, while the NaI concentration in THF solution was 5 mg∙mL-1. The samples were applied on the MALDI steel plate using the dried droplet method: 20 μL of matrix solution was mixed with 1 μL of NaI and 2 μL of sample solution and 1 μL from this mixture was deposited on the ground steel plate. The spectra were acquired in the positive reflectron mode, and the laser ionization power was adjusted slightly above the threshold to produce consistent MS signals. The MS calibration was performed using poly(ethylene glycol) standards applied to the MALDI MS target. The MALDI MS/MS fragmentation experiments were performed in LIFT mode using a Bruker standard fragmentation method. All the instrument parameters employed for the MS/MS process are controlled by the standard MS/MS method provided by the instrument producer. The complete isotopic profile of the parent ion was isolated.

Q3: Finally, the authors should provide more detail on how their results could be used to build new supramolecular compounds.

Answer Q 3: We appreciate the reviewer’s remark. The presence of these macrocycles will help us to obtain a stable water dispersion that will be studied by combining the emerging nanopore resistive pulse sensing technique (Np-RPS) with computational modeling, to identify interactions of EDOT∙TMe-βCD and EDOT∙TMe-γCD  as well their polyrotaxanes  with the aerolysin (Ael) nanopore (see reference 6 in the Main Ms). These results will be published in due course as an example of using this present study to build new supramolecular compounds.

Q4: With these revisions, this manuscript could be improved and provide an even better overview of this area of research. For example, it is essential to apply cyclodextrin to drug delivery systems 10.1016/j.molliq.2022.119548 and 10.1016/j.jiec.2021.10.028 or for CO2 capture 10.3390/molecules27217375. Please discuss this in the introduction or conclusion

Answer Q 4: We agree with the reviewer and have now included in the Introduction part these recommended references (see references 12 & 14).

Q5: Scientific sound corrections (not essential, but will help make your manuscript clear to understand):

"The knowledge gained on this complex compound is a first step toward attaining a greater understanding of the properties and subsequently to generate complex functional materials." --> "The knowledge gained on this complex compound is a first step toward understanding the properties better and generating complex functional materials."

Answer Q 5: We agree with the reviewer and have now improved the readability throughout the manuscript. This particular error was corrected as follows:

The insight gained from studying these compounds is a first step towards generating complex functional materials.

Q6: English corrections?

Answer Q 6: We agree with the reviewer’s remarks and have endeavored to improve the English and readability of the text throughout the Ms.

We hope that our answers help to clarify the reviewers’ comments.
